# Overexpression of *Lolium multiflorum LmMYB1* Enhances Drought Tolerance in Transgenic Arabidopsis

**DOI:** 10.3390/ijms242015280

**Published:** 2023-10-18

**Authors:** Qiuxu Liu, Fangyan Wang, Peng Li, Guohui Yu, Xinquan Zhang

**Affiliations:** 1College of Grassland Science and Technology, Sichuan Agricultural University, Chengdu 611130, China; sicauliuqiuxu@163.com (Q.L.); wfy_zr@163.com (F.W.); lipeng_plant@163.com (P.L.); 2Institute of Agricultural Resources and Environment, Sichuan Academy of Agricultural Sciences, Chengdu 610066, China

**Keywords:** *Lolium multiflorum*, drought, *LmMYB1*, stomatal density

## Abstract

*Lolium multiflorum* is one of the world-famous forage grasses with rich biomass, fast growth rate and good nutritional quality. However, its growth and forage yield are often affected by drought, which is a major natural disaster all over the world. MYB transcription factors have some specific roles in response to drought stress, such as regulation of stomatal development and density, control of cell wall and root development. However, the biological function of MYB in *L. multiflorum* remains unclear. Previously, we elucidated the role of *LmMYB1* in enhancing osmotic stress resistance in *Saccharomyces cerevisiae*. Here, this study elucidates the biological function of *LmMYB1* in enhancing plant drought tolerance through an ABA-dependent pathway involving the regulation of cell wall development and stomatal density. After drought stress and ABA stress, the expression of *LmMYB1* in *L. multiflorum* was significantly increased. Overexpression of *LmMYB1* increased the survival rate of *Arabidopsis thaliana* under drought stress. Under drought conditions, expression levels of drought-responsive genes such as *AtRD22, AtRAB* and *AtAREB* were up-regulated in OE compared with those in WT. Further observation showed that the stomatal density of OE was reduced, which was associated with the up-regulated expression of cell wall-related pathway genes in the RNA-Seq results. In conclusion, this study confirmed the biological function of *LmMYB*1 in improving drought tolerance by mediating cell wall development through the ABA-dependent pathway and thereby affecting stomatal density.

## 1. Introduction

Drought, as a widespread environmental factor in nature [1,2,3], has become one of the most critical factors restricting the yield of forage grass [4,5]. *Lolium multiflorum* is a world-famous cold season forage grass, which has the advantages of fast growth, rich biomass and high nutritional value. It has become the main forage grass for green feed production in winter [6,7]. However, its growth and development are often affected by drought, especially seasonal drought in winter. The results of studies on materials with extreme differences in drought tolerance of *L. multiflorum* show that the metabolic pathways such as sucrose synthesis, root development and ABA signal transduction are involved in the response of drought stress [8,9], and the gene family of MYB, NAC and WRKY is also a response to drought stress [10,11]. The changes in *L. multiflorum* under drought stress are well understood, but the mechanisms by which key drought tolerance genes regulate these phenotypic changes remain unclear.

Plants respond to drought stress in a variety of ways. In order to cope with oxidative stress caused by drought stress, their own enzymatic scavenging system will be used for defense [12,13]. On the other hand, changes in leaf structure and morphology also play an important role in coping with drought. The stomata of plants make them more adaptable to the changeable climate and environment [13,14,15]. Studies have shown that xerophytes improve their adaptability to water deficit by sinking the stomatal surface of the leaf epidermis under drought conditions, increasing stomatal density and decreasing stomatal diameter [16,17]. However, some scholars have come to a different conclusion that the smaller the stomatal density, the stronger the drought resistance, because when the stomatal density decreases, the gas exchange resistance increases, leading to a reduction in transpiration and finally the improvement of drought resistance [18].

Transcription factors (TF) play an important role in plant stress response [19,20]. In our previous study, we found that many TF families, such as MYB, NAC, etc., are involved in drought response of *L. multiflorum* under drought stress (Appendix A) [11]. MYB, as one of the most active TF families in the leaf tissue of *L. multiflorum* under drought stress, has a variety of functions [21]. The conserved domain of the MYB gene family is generally located in the N-terminal and consists of the conserved helix-turn-helix protein–DNA binding region (W-X_19_-W-X_19_-W and F/I-X_18_-W-X_18_-W) consisting of 51–52 amino acids [22].

The MYB gene family regulates cell wall, root and stomatal development [23]. For example, *AtMYB46* and *RrMYB18* promote the biosynthesis of the cell wall by up-regulating the expression of cell wall-associated genes [24]; *AtMYB96* as ABA signal response genes, can activate cuticular wax biosynthesis [25,26]; *AtMYB41* can direct ectopic suberin synthesis [27]; and *MdSIMYB1* regulates the expression of auxin response genes to promote root growth [28]. It was found that *AtMYB60*, as a transcriptional suppressor, was involved in regulating the development of guard cells and stomatal movement. After the knockout of *AtMYB60*, it was found that stomatal size was reduced and drought tolerance was enhanced [29]. Interestingly, *AtMYB61* regulated stomatal size in contrast to *AtMYB60*, and overexpression of *AtMYB61* reduced stomatal size [30]. For MYB to regulate stomatal development, only two genes were found. *AtMYB124* and *AtMYB88* negatively control the expression of genes associated with stomatal development, but positively regulate the expression of genes related to stress conditions [31,32]. However, it is still unclear how the MYB gene family enhances stress resistance in *L. multiflorum*.

Our research group has previously analyzed the MYB gene family of *L. multiflorum*, and conducted preliminary verification of candidate genes through subcellular localization and overexpression verification of *S. cerevisiae*, and found that *LmMYB1* is located in the nucleus and can improve the resistance of *S. cerevisiae* to osmotic stress [11]. In order to further study the physiological and molecular mechanisms of *LmMYB1* under drought stress, this study overexpressed *LmMYB1* in *A. thaliana*. The results showed that overexpression of *LmMYB1* improved the drought tolerance of transgenic *A. thaliana* plants by participating in cell wall biological processes and reducing stomatal density.

## 2. Results

### 2.1. Bioinformatics Analysis of LmMYB1

In this study, the *LmMYB1* gene was cloned from *L. multiflorum* cv. ‘Chuannong No.1′. The full length of the CDS spanned 744 bp coding 248 amino acids. The molecular weight of the protein was 27.98 kDa, and the theoretical isoelectric point was 9.04. The instability index (II) was computed to be 48.66. This classifies the protein as unstable. The LmMYB1 protein contains two typically conserved domains (W-X_19_-W-X_19_-W and F/I-X_18_-W-X_18_-W) at the N-terminal [19], and is an R2R3-MYB transcription factor (Figure 1A). According to previous studies [11], the phylogenetic tree of several MYB proteins with close evolutionary relationships found that LmMYB1 and LpMYB1 (98.10%) had the closest homology relationship, followed by LrMYB (95.62%), TaMYB1 (81.78%), BeMYB140 (85.60%), CpMYB4 (83.20%), ZmMYB31 (75.00%) and AtMYB4 (55.94%) (Figure 1B,C).

### 2.2. Expression Profile of LmMYB1 Induced by ABA and Drought Stress

According to the previous transcriptomic study, we found that *LmMYB1* and genes related to the ABA biosynthetic pathway were strongly induced by drought stress in *L. multiflorum* (Figure 2A) [11]. In order to reveal the biological function of *LmMYB1*, we analyzed its expression pattern under drought stress and ABA treatment on ‘Chuannong No.1′. As is shown in Figure 2, *LmMYB1* was up-regulated 3.57 times after 12 h of drought stress, and 3.21 times after 24 h of ABA treatment, respectively (Figure 2). Altogether, *LmMYB1* potentially responds to drought and ABA. Thus, *LmMYB1* is selected as a candidate gene for further functional studies.

### 2.3. Overexpression of LmMYB1 Could Enhance Abiotic Stress Tolerance

In order to study the function of *LmMYB1* under drought stress, we constructed *LmMYB1* overexpressed Arabidopsis driven by the 35S promoter (Appendix A) and converted it to Arabidopsis by *Agrobacterium* infection. Then, homozygous lines of OE1–OE9 were obtained by hygromycin screening and self-cross for three generations. Among them, the relative expression level of *LmMYB1* in OE1, OE4 and OE9 was the highest, which was about 8, 11 and 10 times higher than that of the lowest lines (Appendix A). Therefore, these three lines were selected for subsequent experiments.

To further verify the function of this gene, Arabidopsis seedlings of similar sizes, WT, OE1, OE4, and OE9, were grown on 1/2 MS plates under different conditions. After one week, there was no significant difference between WT and OE plants on the normal condition (control) plate (Figure 3A). However, under drought stress (200 mM mannitol) and salt stress (100 mM NaCl) conditions, the OE plants exhibited significantly richer above-ground fresh weight than the WT (Figure 3B). Under drought stress (200 mM mannitol) and ABA stress (5 μM), the OE plants showed significantly longer root length than the WT (Figure 3C). In conclusion, exogenous expression of *LmMYB1* can improve the tolerance of Arabidopsis seedlings to drought stress, salt stress and ABA stress.

### 2.4. LmMYB1-Activated ROS Clearing System Enhances Drought Tolerance

To further study the resistance of OE Arabidopsis adult plants to drought stress, a potted natural drought experiment was conducted in this study. The results showed that, after 14 days without water, the WT rosette leaves turned yellow and wilted, and some died. Some rosette leaves of the OE lines turned yellow but most of them were only restricted in growth (Figure 4A). After resuming watering, it was observed that most of the OE lines recovered, while only a small part of the WT did (Figure 4A,B). Under drought stress, on the one hand, the OE lines have significantly lower MDA content and electrolyte leakage (EL) (Figure 4C,D), indicating that WT plant cells are more seriously damaged by drought stress. On the other hand, the OE lines showed significantly more active antioxidant enzyme activity, indicating that the OE strain had a stronger ability to clear the negative effects of drought stress (Figure 4E–G). Taken together, overexpression of *LmMYB1* can reduce oxidative damage under drought by reducing MDA, EL and increasing antioxidant enzyme activity of Arabidopsis adult plants.

### 2.5. LmMYB1 Enhances the Expression of Drought-Related Genes

Drought response involves multiple pathways, such as the proline synthesis pathway, ABA dependent pathway, dehydration response pathway, etc. Many genes are involved in these processes, and these genes often act as marker genes to detect whether the pathway is responsive. To analyze the effect of *LmMYB1* overexpression on Arabidopsis stress-response genes, the expression levels of six genes (*AtRAB18, AtAREB, AtRD22, AtCOR47, AtCOR15A* and *AtDREB*) were determined after drought using qRT-PCR. As shown in Figure 5, all genes were up-regulated under drought stress. Meanwhile, under drought stress, the expression level of ABA-dependent-pathway-related genes (*AtRAB18, AtAREB* and *AtRD22*) in overexpressed *LmMYB1* lines was significantly higher (*p* < 0.05) than that in the WT after drought stress, suggesting that *LmMYB1* may be dependent on the ABA-dependent pathway for drought response (Figure 5). Stress-related genes (*AtCOR47, AtCOR15A* and *AtDREB*) were also significantly up-regulated, suggesting that *LmMYB1* may regulate plant response to drought stress through the pathways corresponding to these genes.

### 2.6. Transcriptomic Analysis of OE vs. WT under Drought Stress

In order to explore the molecular mechanism of overexpression of *LmMYB1* to improve drought resistance, RNA-Seq analysis was performed on transgenic and wild-type *A. thaliana* under drought stress. Compared with the WT, 143 genes were up-regulated and 195 genes down-regulated in *LmMYB1-OE* plants (Appendix A). The term of up-regulated gene enrichment is mostly related to cell wall pathways, such as the Glucuronoxylan metabolic process, Xylan biosynthetic process, Xylan metabolic process, cell wall macromolecule biosynthetic process, etc., which specifies the stronger activity of genes related to cell wall pathways in *LmMYB1-OE* plants than the WT (Figure 6A,C). Most of the terms in which down-regulated expression genes were enriched were related to responses to other stresses (Figure 6B,C). The results showed that, under drought stress, overexpression of *LmMYB1* in *A. thaliana* tended to mobilize cell wall-related pathways in response to drought, while reducing the response to other stresses, which should be a strategy for *LmMYB1* to cope with drought stress as an environmental stress factor.

To confirm the RNA-Seq results, 11 DEGs from different categories were picked for qRT-PCR analysis (Figure 6D). Among the five cell wall pathway-related genes (*PRR1, IRX12, CTL2, KIN1* and *XTH6*), except *XTH6*, all the others were up-regulated in *LmMYB1-OE* lines under drought stress. Similarly, the expression of antioxidant oxidase and JA signal response-related genes were also up-regulated in *LmMYB1-OEs* in response to drought treatment. Taken together, these results strongly demonstrated that *LmMYB1* mediated drought tolerance by affecting cell way-related pathways, antioxidant oxidase activity and JA signal response.

To further explore the phenotypic effects of the genetic changes, the epidermis of the WT and OE lines were observed by optical microscope. Three biological replicates were set for the WT and OE lines, and fixed sections of rosette leaves in the same position were selected for each replicate. For each section, 10 photos of visual fields were randomly collected under visual fields (10 × 40) to calculate stomatal density. Interestingly, the results showed that the average stomatal density of the WT, OE1 and OE4 per field was 6.267, 5.200 and 4.933. OE lines significantly decreased leaf stomatal density by 17.02% (*n* = 30, *p* = 0.0026) and 21.29% (*n* = 30, *p* = 0.0002) compared with the WT (Figure 7). These results further suggest that *LmMYB1* could enhance drought tolerance of *A. thaliana* by reducing stomatal density. Unfortunately, analysis of the transcriptome data found no significant differences in genes related to stomatal development and its signal transduction regulatory pathways between WT and OE (Appendix A).

## 3. Discussion

Drought tolerance is a complex and inherent trait of plants that responds to drought stress at multiple levels [33]. At one level, changes in transcriptional regulation involve thousands of genes, and transcription factors play a pivotal role in this process [34]. The MYB family is one of the largest transcription factor families and shows specific roles in response to abiotic stress [23,35,36]. This study systematically characterized the molecular functions of *LmMYB1* in *A. thaliana* in response to drought stress. As a transcription factor, LmMYB1 increases the expression of downstream stress-related genes and ABA-dependent genes. At the same time, *LmMYB1* improves ROS clearing by increasing antioxidant enzyme activity, and improves osmoprotection by decreasing MDA and EL. Further, we observed that *LmMYB1* caused a decrease in the epidermal stomatal density of Arabidopsis. The results provided the foundation for further clarification of the stress tolerance mechanism and molecular breeding of *L. multiflorum*.

As an important plant hormone, abscisic acid (ABA) plays a key role in regulating plant responses to abiotic stresses [37]. Studies have shown that ABA is involved in plant responses to various stresses such as drought, salt, and high temperature [38]. This study was studied the *LmMYB1* from *L. multiflorum*, which was found to be in the HTH conservative structure domain, which was induced by drought and ABA in *L. multiflorum* (Figure 2). At the same time, the drought stress treatment increased the expression of *AtRD22*, *AtRAB18* and *AtAREB* in the transgenic *LmMYB1 Arabidopsis* (Figure 5). The root growth of the transgenic *Arabidopsis* seedlings was significantly better than the wild type (Figure 3A,C). These results show that *LmMYB1* is involved in the plant’s response to adversity through the ABA-dependent pathway (Figure 8).

As sessile organisms, plants cannot move and could make some morphological changes in the face of external stresses to resist them [39]. Changes in leaf morphology include changes in leaf shape, epidermis and stomata [40,41,42]. Based on the statistics of stomatal parameters and drought phenotypes of rice populations after radiation mutagenesis, researchers found that smaller stomata with lower stomatal density had higher resistance and higher seed yield under drought stress [43]. Similar patterns have been found in grape [44] and carrot [45], where stomatal density is negatively correlated with plant drought tolerance. The findings in this study, that *LmMYB1* overexpressed Arabidopsis has lower stomatal density and higher drought resistance, are also consistent with these studies just mentioned (Figure 7). As stomata are exchange channels for water and gas, it seems like a paradoxical and interesting question that overexpression of *LmMYB1* reduces stomatal density conferring drought tolerance with little influence on normal growth (Figure 3B) rate. Since *L. multiflorum* is a kind of high-quality forage, *LmMYB1* would be a valuable candidate gene conferring drought tolerance. We will validate its function in its native plants via overexpression assay and investigate detailed information and the mechanisms of *LmMYB1*-mediated stomatal density in the regulation of drought tolerance in future studies.

Stomatal density is affected by transcription factors, signal peptides and environmental factors. These internal and external factors can not only affect stomatal development independently, but also synergistically regulate the formation of stomata [46]. The regulation process of stomatal density and distribution on the leaf surface is highly plastic, and there are many influencing factors [47,48,49]. In addition to the identified genes directly regulating stomatal development (SPCH/MUTE/FAMA) [50], other potential factors may also contribute. Studies have shown that fucosylation-dependent dimerization of the cell wall pectic domain rhamnogalacturonan-II may be essential for stomatal development and leaf water retention [51]. In addition, it has been found that galactosyltransferase regulates the synthesis and composition of pectin, thereby affecting stomatal development and dynamics [52]. In this study, the RNA-Seq data revealed a significant enrichment of a large number of up-regulated genes related to cell wall biological pathways (Figure 6A,B), such as genes involved in the regulation of secondary cell wall biosynthesis, pinoresinol reductase catalyzes (PRR1) [53], irregular xylem12 (IRX12)[54] and chitinase-like2 protein (CTL2) [55], which play important roles in cell wall development. A recent study found that *ZmIRX15A* (irregular xylem 15A), a gene encoding the xylem deposition enzyme in maize, negatively regulates stomatal density by changing cell wall composition [56], which is consistent with our observed results (Figure 6D). However, how *LmMYB1* mediates downstream genes integrated in cell wall-related pathways in regulating drought tolerance remains unclear. Further studies on such themes would provide more knowledge of the MYB-regulated network. 

## 4. Materials and Methods

### 4.1. Lolium Multiflorum Material, Growth Condition and Treatment

The samples of *L. multiflorum* cv. ‘Chuannong No.1′ for expression pattern analysis were planted in the growth chamber at Chengdu Campus of Sichuan Agricultural University. The diurnal cycle of the growth chamber was 14/10 h and 22/20 °C. Photons were 750 μmol m^−2^·s^−1^. The air humidity was 70%. The seeds of *L. multiflorum* cv. ‘Chuannong No.1′ were disinfected and cultured in square pots (height 6 cm, length 25 cm, width 6 cm). The seeds were disinfected with 1% sodium hypochlorite solution for 10 min, washed with deionized water 3–5 times, and 150 seeds were evenly sown in each square basin. Water was used before germination, and 1/2 Hoagland solution was used after germination. When the seedlings were grown under normal conditions for 30 days, stress treatment was carried out. To simulate drought treatment, 15% PEG-6000 was used, and 30 μmol ABA was used to stimulate ABA treatment. Leaves were collected at 0 h, 3 h, 6 h, 12 h and 24 h after drought and ABA treatment. All tissues were stored in liquid nitrogen immediately after sampling and then at −80 °C.

### 4.2. Multiple Sequence Alignment and Bioinformation Analysis of the LmMYB1 Gene

The *LmMYB1* gene sequence was extracted from transcriptome data [11], after cloning, to obtain the exact sequence. At the same time, other sequences of homologous proteins were downloaded from the National Center for Biotechnology Information (NCBI). DNAMAN software, version 5.1.5 (Lynnon BioSoft, San Ramon, CA, USA), was used to analyze the amino acid sequences under default parameters. A phylogenetic tree was subsequently constructed using the neighbor-joining method with MEGA software (version 11.0.10) with the following parameters: the bootstrap test method (1000 replicates) and pairwise deletion [57].

### 4.3. Generation of Transgenic Arabidopsis Plants

For Arabidopsis transformation, the CDS of *LmMYB1* was cloned from the cDNA of *L. multiflorum* leaves and ligated into the PHG vector under the control of the CaMV 35S promoter. PHG-LmMYB1 was further transformed into *Agrobacterium tumefaciens* GV3101. Arabidopsis transformation was conducted with the floral dipping method with *A. thaliana* Col-0 [58]. The positive transgenic plants were screened by Hygromycin, and at least 9 independent transgenic plants were obtained until the T3 generation for further analysis.

### 4.4. Evaluation of Drought Resistance in Transgenic Lines

The seeds of different lines were seeded in 1/2 MS medium and cultured vertically. When the root length was about 1 cm, they were transferred to 1/2 MS, 1/2 MS + 5 μM ABA, 1/2 MS + 200 mM mannitol and 1/2 MS + 100 mM NaCl in a sterile environment. The phenotypes were observed and photographed after stress, while above weight and root length of the seedlings were measured.

The WT and OE lines (OE1, OE4, OE9) with the same growth after 12 days of germination on 1/2 MS medium were transferred to a soil culture pot (the substrate was peat soil, perlite and vermiculite, the volume ratio was 8:1:1). After 3 weeks of cultivation under normal conditions, watering was stopped, and the phenotype was observed and photographed when the natural drought occurred. One week after rewatering, observation continued and photographs were taken again, and the survival rate was recorded.

Drought stress-related physiological indicators including antioxidant enzymes (POD, SOD and CAT), MDA content and electrolyte leakage (EL) were measured before and after drought stress to identify the responses of different wild type and overexpression lines to drought stress.

### 4.5. Extraction of RNA and qRT-PCR Analysis

Total RNA was extracted according to the manufacturer’s instructions for the Trelief^®^ RNAprep Pure Plant Kit (# TSP411, Tsingke Biotechnology Co., Ltd., Beijing, China). For the next step of reverse transcription to proceed smoothly, the RNA concentration should be above 200 ng/uL, and the OD260/OD280 value should be 1.8–2.0. Then, the first-strand cDNA was synthesized from 1000 ng RNA using SynScript^®^ Ⅲ RT SuperMix for qPCR (#TSK314S, Tsingke Biotechnology Co., Ltd., Beijing, China). qRT-PCR was performed using an ArtiCan^CEO^ SYBR qPCR Mix (#TSE401, Tsingke Biotechnology Co., Ltd., Beijing, China). The reaction solutions and qRT-PCR procedures were performed according to the kit’s instructions. Finally, the qRT-PCR reaction was performed using CFX Connect™ Real-Time System (Bio-Rad, Hercules, CA, USA). *LmHIS3* and *LmeIF4A* (Appendix A) were used as the reference genes to analyze the expression patterns of *LmMYB1* under drought and ABA in *L. multiflorum* [59]. The specific primers used for the expressions of stress-related and ABA-signaling gene analysis are shown in Appendix A. *AtACT2* (Appendix A) was used as the reference gene to analyze expression levels of marker genes. qRT-PCR data were analyzed using the 2^−∆∆Ct^ method [60]. Three technical replicates of each sample were analyzed with qRT-PCR.

### 4.6. RNA-Seq Analysis

The aboveground leaf of the wild type and overexpressed line (OE4) of *A. thaliana* were taken after 30 days of growth by natural drought after 6 days. Each sample consisted of three individual plants, with three biological replicates at each point. Total RNA was extracted for cDNA synthesis. The construction of the cDNA library, sequencing, and data filtering were performed by Gene Denovo (Gene Denovo Co., Ltd., Guangzhou, China). The differentially expressed genes (DEGs) were analyzed using DESeq2 (log2 fold changes ≥ 2, FDR < 0.001) [61]. To understand which genes *LmMYB1* affects and what function those genes have, enrichment analyses and gene ontology (GO) annotation were analyzed using TopGO and the online tool agriGO (http://bioinfo.cau.edu.cn/agriGO/, accessed on 15 September 2021) [62]. To cluster and screen DEGs and determine which pathways are affected by *LmMYB1*, Kyoto Encyclopedia of Genes and Genomes (KEGG) pathways were analyzed by cluster-Profiler [63]. All the raw data generated are housed in the National Genome Sciences Data Center (CRA011179).

### 4.7. Measurement of Stomatal Density

Samples were taken at 30 d from Arabidopsis leaves to observe stomatal density. The clipped leaves were quickly placed in 2.5% glutaraldehyde (pH 7.2) for fixation. The largest leaf of 3 independent plants from each line was taken and stored in a refrigerator at 4 °C. The leaves were removed from the glutaraldehyde fixing solution, the leaf skin was wiped clean with absorbent cotton dipped in alcohol and a thin layer of nail polish was smeared on the skin. After drying, the nail polish film was peeled off with tweezers and placed on a slide so that it could be stretched naturally and observed under a microscope. A slice was made for each sample. Ten visual fields were randomly selected from the section, observed under an optical microscope (10 × 40), photographed, and the number of visual fields was counted and the average value calculated.

### 4.8. Statistical Analysis

In this study, the values obtained in the figures were expressed as the means. The error bars were based on standard deviation (SD) in figures. A one-way ANOVA followed by Tukey’s multiple comparisons tests were used to determine the significant differences among groups.

## 5. Conclusions

In this study, the coding sequence of the *LmMYB1* gene was cloned from *L. multiflorum* and functionally verified in Arabidopsis overexpression. The results showed that *LmMYB1* positively regulated drought response in transgenic Arabidopsis, such as drought phenotype, antioxidant enzyme activity and MDA content. In addition, the up-regulation of several ABA-related drought stress genes indicated that *LmMYB1* played a positive role in the ABA-dependent pathway. The reduction in stomatal density and changes in a large number of genes involved in cell wall biological pathways suggest pathways through which *LmMYB1* may improve plant drought tolerance. Taken together, *LmMYB1* acts as an ABA-dependent transcription factor and positively regulates the drought response of transgenic Arabidopsis by reducing stomatal density and up-regulating the expression of drought stress-related genes and cell wall biological pathway-related genes, which provides a basis for further understanding the drought response mechanism of *L. multiflorum*.

## Figures and Tables

**Figure 1 ijms-24-15280-f001:**
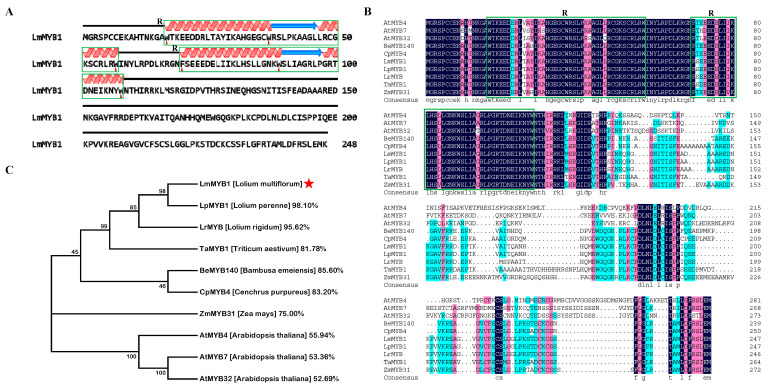
Multi-sequence alignment and evolutionary relationship analysis of the LmMYB1 protein. (**A**) Secondary structure of the LmMYB1 protein. The green boxes represent conserved domains, and the red triangles represent conserved amino acids. (**B**) Comparison between the homology of the LmMYB1 protein and the MYB protein in other plants. The conserved regions of the amino acid sequence are marked with green boxes. R represents the conserved domain of MYB (W-X_19_-W-X_19_-W and F/I-X_18_-W-X_18_-W). The black background represents a 100% consistent sequence. Red and blue represent 75% and 50% respectively. (**C**) Phylogenetic tree analysis of LmMYB1 (noted with a red star) and homologous in other plants.

**Figure 2 ijms-24-15280-f002:**
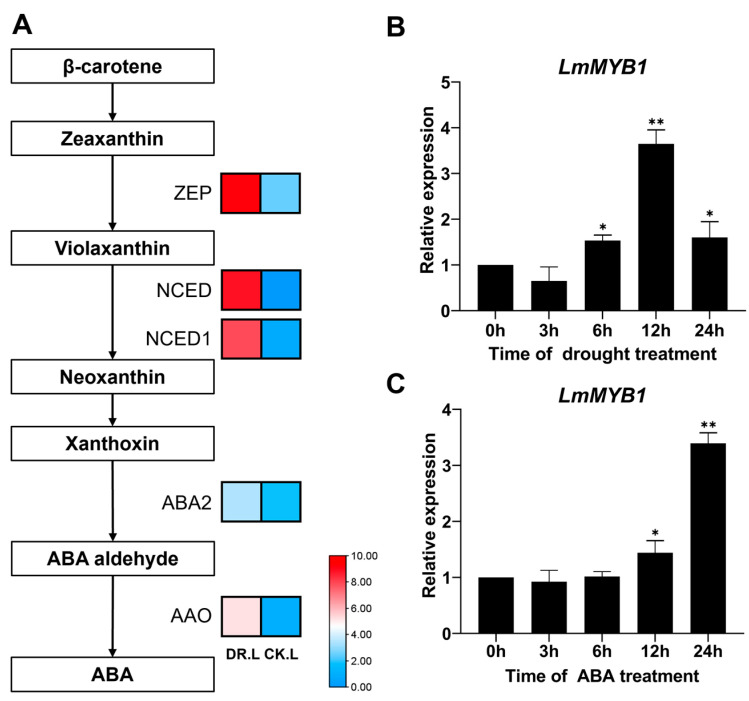
Gene expression profile of *L. multiflorum* under abiotic stress. (**A**) Related genes of the ABA biosynthetic pathway and *LmMYB1* under drought stress in the previous transcriptome data [11]. The ordinate represents the fold change (CK vs. DR) of genes under drought stress. There are three biological replicates per treatment, with the average shown in the figure. The color from blue to red indicates low to high gene expression. The expression profile of *LmMYB1’s* response to drought stress (**B**) and ABA treatment (**C**) in ‘Chuannong No. 1′. Three biological repetitions per time point, three technical repetitions. The standard deviation is represented by the error bar. *LmHIS3* and *LmeIF4A* were used as the reference genes and qRT-PCR data were analyzed using the 2^−∆∆Ct^ method. The asterisk indicates the difference compared to 0 h (* *p* < 0.05, ** *p* < 0.01).

**Figure 3 ijms-24-15280-f003:**
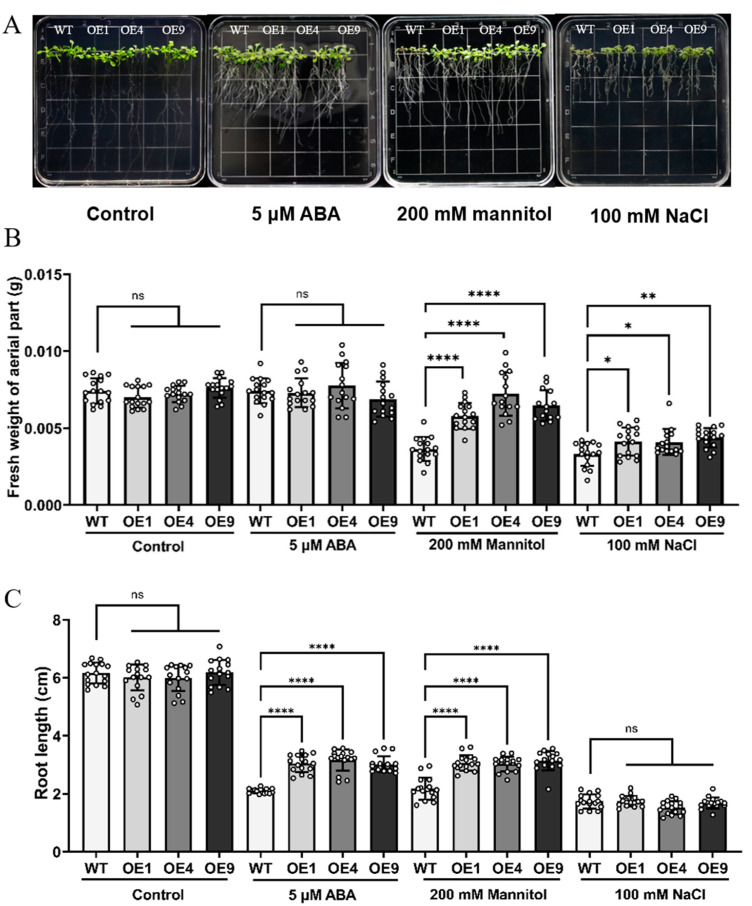
Comparison of abiotic stress tolerance of overexpressing *LmMYB1* lines with wild type on 1/2 MS medium. (**A**) Phenotype of the WT and OE lines under 5 μM ABA, 200 mM mannitol and 100 mM NaCl compared to 1/2 MS. (**B**,**C**) Fresh weight and root length of the WT and OE lines under different treatments, *n* = 15, * *p* < 0.05, ** *p* < 0.01, **** *p* < 0.0001, ns, non-significant.

**Figure 4 ijms-24-15280-f004:**
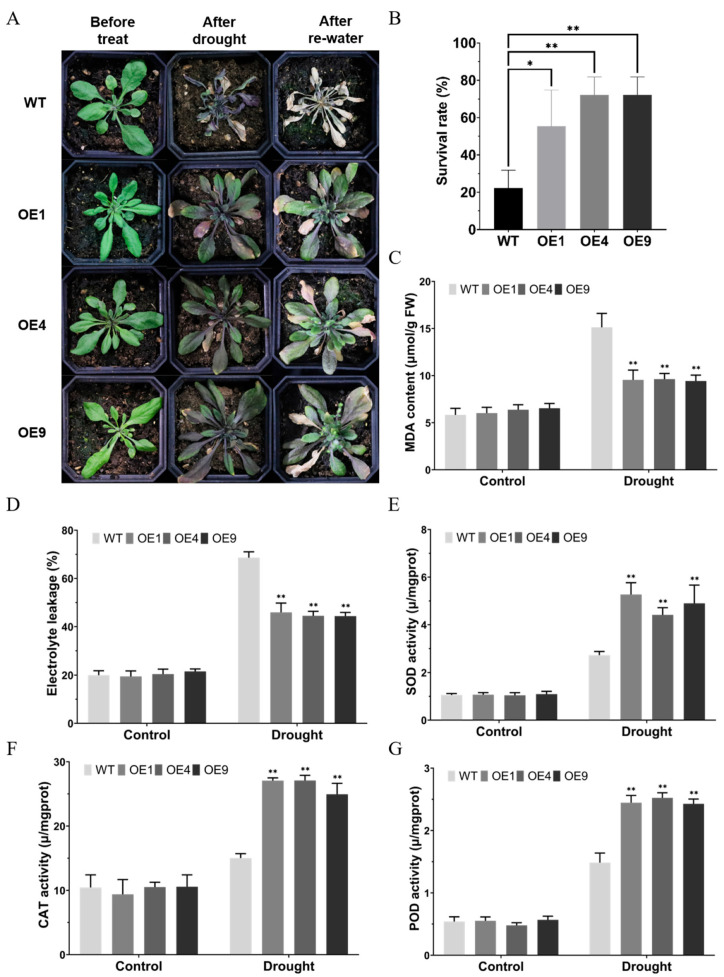
Overexpression of *LmMYB1* can improve drought tolerance in *A. thaliana*. (**A**) Phenotype of the WT and transgenic lines (OE1, OE4, OE9) under drought stress. (**B**) Survival rate of the WT and transgenic lines after re-watering. (**C**–**G**) Shows measurements of tolerance-related physiological parameters. Data are presented as mean and SD values of four independent experiments. Asterisks indicate significant difference (* *p* < 0.05, ** *p* < 0.01, by independent sample *t*-test) between the WT and OE lines.

**Figure 5 ijms-24-15280-f005:**
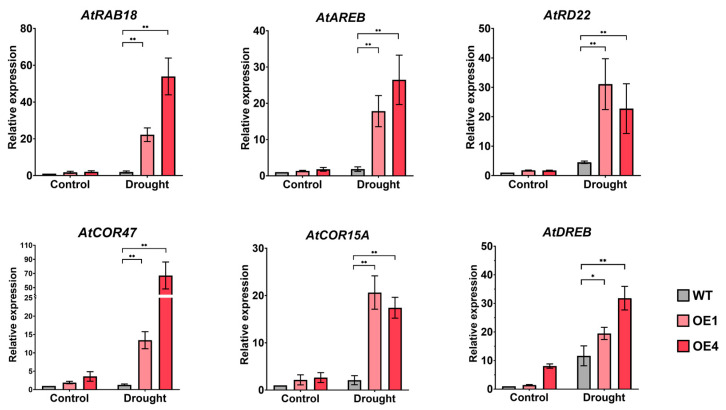
Analysis of stress resistance gene expression in WT and overexpressed *LmMYB1* under drought stress. The value is the average of three biological repetitions and three technical repetitions. The standard deviation is represented by the error bar. *AtACT2* was used as the reference gene and qRT-PCR data were analyzed using the 2^−∆∆Ct^ method. The asterisk indicates the difference compared to WT (* *p* < 0.05, ** *p* < 0.01).

**Figure 6 ijms-24-15280-f006:**
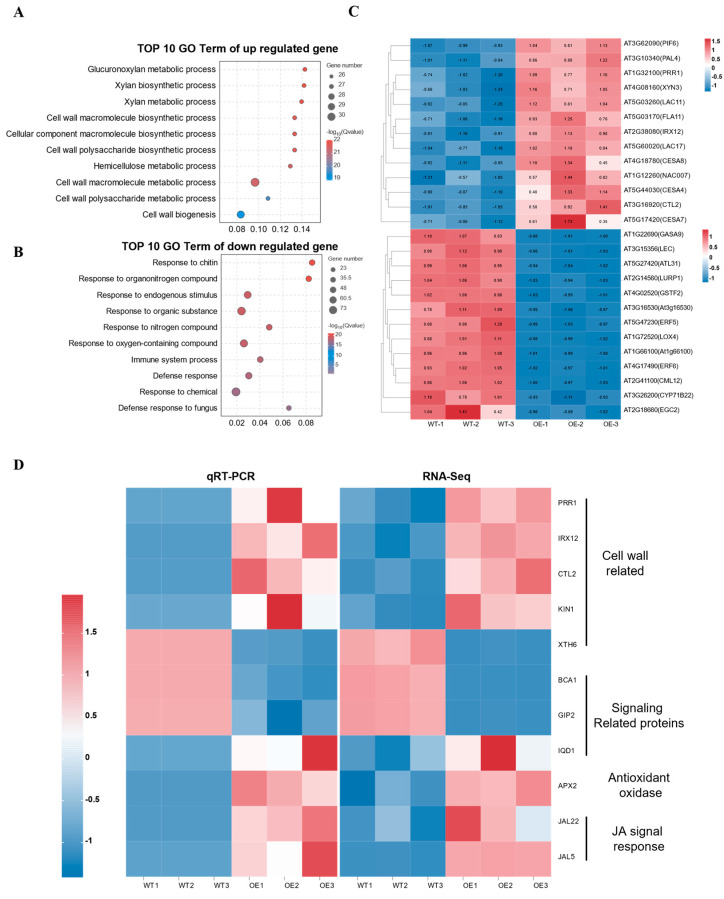
GO enrichment analysis and core DEGs heat map of WT vs. OE gene. (**A**,**B**) The top ten terms enriched by up-regulated and down-regulated genes were shown, respectively. The vertical axis represents different biological processes, and the horizontal axis represents rich factors. The size of the dots represents the number of genes enriched. The color of the dots represents the -log_10_(Qvalue) of the event. (**C**) The heat map of core genes. The core gene is derived from the biological process enriched in the previous step. The right ordinate is the number and name of different genes. (**D**) Validation of transcriptome data of key genes. In qRT-PCR, each square is formed by three technical repetitions and *AtACT2* was used as the reference gene. The right ordinate shows the name of the gene and its approximate functional range. The color from blue to red indicates low to high gene expression.

**Figure 7 ijms-24-15280-f007:**
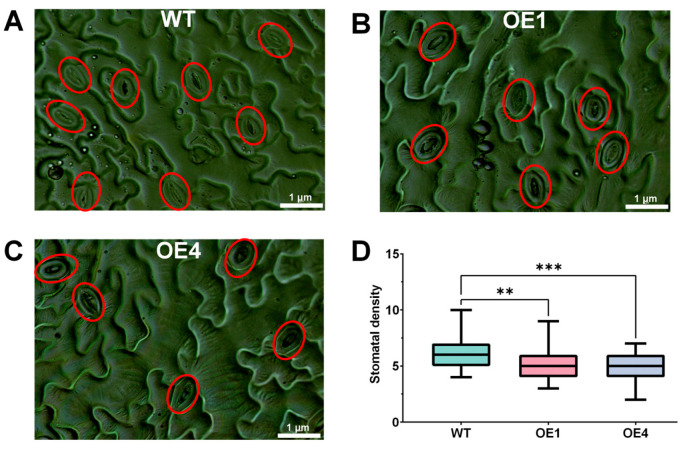
Overexpression of *LmMYB1* decreased stomatal density in *A. thaliana*. (**A**–**C**) Representation figure of stomatal density of the WT and transgenic lines (OE1, OE4). The red circle represents each stoma. Scale = 1 μm. (**D**) Statistics of stomatal density of the WT and transgenic lines (OE1, OE4). There were three biological replicates per line and 10 visual fields (10 × 40) were collected for each biological replicate. Asterisks indicate significant difference (** *p* < 0.01, *** *p* < 0.001, by independent sample *t*-test) between WT and OE lines.

**Figure 8 ijms-24-15280-f008:**
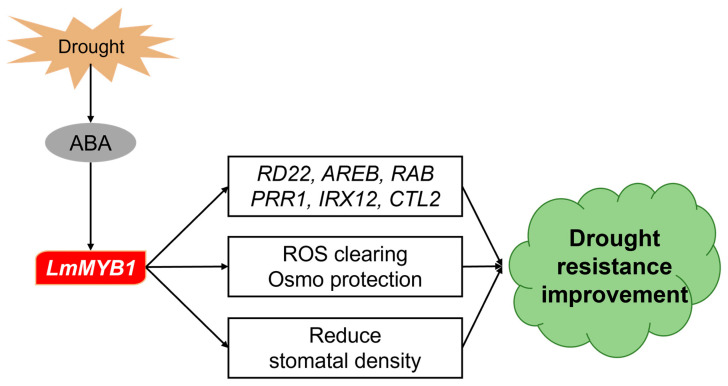
Hypothetical model of the *LmMYB1*-regulated network in the drought response.

## Data Availability

All the transcriptome sequencing raw data are housed in the National Genome Sciences Data Center (CRA011179).

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
