# Peer review of "Overexpression of Lolium multiflorum LmMYB1 Enhances Drought Tolerance in Transgenic Arabidopsis"

_ijms, 2023, doi:10.3390/ijms242015280_

Round 1

Reviewer 1 Report

The manuscript titled “Overexpression of Lolium multiflorum LmMYB1 enhances drought tolerance in transgenic Arabidopsis”

This study reveals that LmMYB1 enhances plant drought tolerance through an ABA-dependent pathway, regulating cell wall development and stomatal density. The authors have observed that overexpression of LmMYB1 increases survival rates in Arabidopsis thaliana under drought stress and up-regulates drought-responsive genes. Overall, the current work was designed and executed well. The reviewer appreciates the effort of the authors to prove their hypothesis using series experiments. The strength of the study is the experimental portions and nevertheless, the reviewer has a few comments regarding this study. Thus, the authors need to consider the following comments to improve the quality of this manuscript.

Gene names should be in italics. Check and revise the same throughout the manuscript. Eg. lines 217, 219, 265, and Fig.8 etc. When the gene name comes along with the short form of plant scientific names, it should be in italics.

Section 2.6 needs to be improved.

Figure 6 description should be improved with details instead of just mentioning A-D represents. It should be self-explanatory. What does the color and scale bar represent? etc. I suggest authors read the recent references to update the figure descriptions.  

In conclusion section, write a few lines about future perspectives or hypotheses about the study. It will be useful to the readers for ease of understanding to design their study related to this studied issue.

Check the space and punctuation errors throughout the manuscript.

Section 4.6 should be improved with details. Authors should mention why you have performed series of bioinformatic analyses after the sequencing. Simply listing the tools in the methods section is not enough.

There is no attachment for the Supplementary files. Please provide that for review and publication purposes.  

Authors should provide the vector construction and overexpression image in the supplementary section for ease of understanding of the readers.

To check the gene/DNA integration and the copy number, Southern Blot hybridization is required, but why authors did not perform the southern blot analysis?

Please describe more about in the RNA extraction section in the materials and method section. What is RNA concentration? How much concentration was used for cDNA conversion.

Authors should provide a separate statistical analysis section in the materials and methods part.  

Author Response

Dear reviewer:

I would like to take this opportunity to express my great appreciation for your time and valuable suggestions. According to your advice, we had revised the relevant part in the manuscript as follows.

Point 1: Gene names should be in italics. Check and revise the same throughout the manuscript. Eg. lines 217, 219, 265, and Fig.8 etc. When the gene name comes along with the short form of plant scientific names, it should be in italics.

Response 1: Thank you for your comment. We have carefully checked and revised such mistakes in the whole manuscript.

Point 2: Section 2.6 needs to be improved.

Response 2: Thanks. The rewrite of section 2.6 is complete. A more in-depth interpretation of the information presented in Figure 6 was added to enrich the manuscript (Ln.199-208).

Point 3: Figure 6 description should be improved with details instead of just mentioning A-D represents. It should be self-explanatory. What does the color and scale bar represent? etc. I suggest authors read the recent references to update the figure descriptions.

Response 3: Thanks. All mentioned have been revised.

Point 4: In conclusion section, write a few lines about future perspectives or hypotheses about the study. It will be useful to the readers for ease of understanding to design their study related to this studied issue.

Response 4: Thanks for your suggestion. The hypotheses and future research directions have been added (Ln. 291-296).

Point 5: Check the space and punctuation errors throughout the manuscript.

Response 5: Thanks. Done.

Point 6: Section 4.6 should be improved with details. Authors should mention why you have performed series of bioinformatic analyses after the sequencing. Simply listing the tools in the methods section is not enough.

Response 6: Thanks. In the revised manuscript, the purpose of the bioinformatics analysis process is explained (Ln. 384-389).

Point 7: There is no attachment for the Supplementary files. Please provide that for review and publication purposes.

Response 7: Thanks. The supplementary files have been added.

Point 8: Authors should provide the vector construction and overexpression image in the supplementary section for ease of understanding of the readers.

Response 8: Thank you for your comment. In order to make it easier for readers to understand, the constructed vector map was added as supplementary file (Fig. S2) in the revised manuscript.

Point 9: To check the gene/DNA integration and the copy number, Southern Blot hybridization is required, but why authors did not perform the southern blot analysis?

Response 9: Thanks for your question. Southern blot is indeed a classic experimental method used to detect the gene/DNA integration and the copy number in more application-oriented studies. Till now, Arabidopsis transformation has been commonly and easily used in molecular biology.  In this study, RT-PCR and qRT-PCR analysis successfully identified all three LmMYB1 overexpression transgenic Arabidopsis lines (OE1, OE4, OE9), and all the three independent OE plants (Fig. S3) showed significantly high expression levels of LmMYB1 and consistent drought tolerance phenotype compared to WT.

Point 10: Please describe more about in the RNA extraction section in the materials and method section. What is RNA concentration? How much concentration was used for cDNA conversion.

Response 10: Thanks. Done (Ln.364-367).

Point 11: Authors should provide a separate statistical analysis section in the materials and methods part.

Response 11: Thanks. Section 5.8 has been added to describe how the statistical analysis was performed in this study.

Reviewer 2 Report

Review of ijms-2632702

Overexpression of Lolium multiflorum LmMYB1 enhances drought tolerance in transgenic Arabidopsis

Qiuxu Liu , Fangyan Wang , Peng Li , Guohui Yu , Xinquan Zhang

The authors wished to study drought resistance in Lolium multiflorum.  They therefore performed RNAseq analysis and found that  expression of Lolium multiflorum LmMYB1 increased under drought conditions.  They therefore cloned the cds of Lolium multiflorum LmMYB1 and transformed it into Arabidopsis under the control of the 35S promoter.  They then compared various types of responses of OE and wt plants to drought, and concluded that over-expression of LmMYB1 enhanced drought tolerance of Arabidopsis. One mechanism was via reducing the stomatal density.

Overall, it is a solid study that seems to have been competently performed. However, I have concerns about the replication.

In figure 2 it is unclear how many replicates were in each test, and how many times the experiment was repeated.  In contrast, this is very clear in figures 3. 4

In all captions to figures showing RT-qPCR experiments please list the numbers of biological replicates, numbers of technical replicates and the reference gene used.

Why do 5. Conclusions come before 4. M&M?

Please rewrite lines 116-118 for clarity

Please rewrite lines 157-158 for clarity

Please rewrite lines 176-179 for clarity

Please rewrite lines 291-292 for clarity

Please rewrite lines 357-363 in past tense.

The English is good, but there are numerous minor mistakes.

Author Response

Dear reviewer:

I would like to take this opportunity to express my great appreciation for your time and valuable suggestions. According to your advice, we had revised the relevant part in the manuscript as follows.

Point 1: In figure 2 it is unclear how many replicates were in each test, and how many times the experiment was repeated. In contrast, this is very clear in figures 3. 4.

Response 1: Thanks for your suggestion. We have added the detailed parameters of this part of the experiment in the caption of the figure to make it easier to understand (Ln. 113-119).

Point 2: In all captions to figures showing RT-qPCR experiments please list the numbers of biological replicates, numbers of technical replicates and the reference gene used.

Response 2: Thanks. The information has been added to Fig. 2, Fig. 5, Fig. 6.

Point 3: Why do 5. Conclusions come before 4. M&M?

Response 3: Thanks. We have checked and updated it.

Point 4: Please rewrite lines 116-118 for clarity. Please rewrite lines 157-158 for clarity. Please rewrite lines 176-179 for clarity. Please rewrite lines 291-292 for clarity. Please rewrite lines 357-363 in past tense.

Response 4: Thank you for your suggestion. We have revised these sentences in more detail displayed in Ln. 122-128, Ln. 166-168, Ln. 192-206, Ln. 352-329 and Ln. 396-400 respectively.

Round 2

Reviewer 1 Report

Authors have incorporated all my suggestion in the revised manuscript. Current version of the manuscript seems better than previous version. Therefore, I recommend this manuscript can be accepted for Publication in IJMS